# Real-time two-axis control of a spin qubit

Fabrizio Berritta [1] ✉, Torbjørn Rasmussen [1], Jan A. Krzywda[2], Joost van der Heijden[3], Federico Fedele [1], Saeed Fallahi[4,5], Geoffrey C. Gardner [5], Michael J. Manfra [4,5,6,7], Evert van Nieuwenburg[2], Jeroen Danon [8], Anasua Chatterjee [1] ✉ & Ferdinand Kuemmeth [1,3] ✉

Optimal control of qubits requires the ability to adapt continuously to their ever-changing environment. We demonstrate a real-time control protocol for a two-electron singlet-triplet qubit with two fluctuating Hamiltonian parameters. Our approach leverages single-shot readout classification and dynamic waveform generation, allowing full Hamiltonian estimation to dynamically stabilize and optimize the qubit performance. Powered by a field-programmable gate array (FPGA), the quantum control electronics estimates the Overhauser field gradient between the two electrons in real time, enabling controlled Overhauser-driven spin rotations and thus bypassing the need for micromagnets or nuclear polarization protocols. It also estimates the exchange interaction between the two electrons and adjusts their detuning, resulting in extended coherence of Hadamard rotations when correcting for fluctuations of both qubit axes. Our study highlights the role of feedback in enhancing the performance and stability of quantum devices affected by quasistatic noise.

Feedback is essential for stabilizing quantum devices and improving their performance. Real-time monitoring and control of quantum systems allows for precise manipulation of their quantum states[1,2]. In this way, it can help mitigate the effects of quantum decoherence and extend the lifetime of quantum systems for quantum computing and quantum sensing applications[3], for example in superconducting qubits[4–8], spins in diamond[9–14], trapped atoms[15,16], and other platforms[17–22].

Among the various quantum-information processing platforms, semiconductor spin qubits[23,24] are promising for quantum computing because of their long coherence times[25] and foundry compatibility[26]. Focusing on spin qubits hosted in gate-controlled quantum dots (QDs), two-qubit gate fidelities of 99.5% and single-qubit gate fidelities of 99.8% have recently been achieved in silicon[27]. In germanium, a four-qubit quantum processor based on hole spins enabled all-electric qubit logic and the generation of a four-qubit Greenberger-Horne-Zeilinger state[28]. In gallium arsenide, simultaneous coherent exchange

rotations and four-qubit measurements in a 2 × 2 array of singlet-triplets were demonstrated without feedback, revealing site-specific fluctuations of nuclear spin polarizations[29]. In silicon, a six-qubit processor was operated with high fidelities enabling universal operation, reliable state preparation and measurement[30].

Achieving precise control of gated qubits can be challenging due to their sensitivity to environmental fluctuations, making feedback necessary to stabilize and optimize their performance. Because feedback-based corrections must be performed within the correlation time of the relevant fluctuations, real-time control is essential. Continuous feedback then allows to calibrate the qubit environment and to tune the qubit in real time to maintain high-fidelity gates and improved coherence, for instance by suppressing low-frequency noise and improving $\pi$-flip gate fidelity[31]. An active reset of a silicon spin qubit using feedback control was demonstrated based on quantum non-demolition readout[32]. Real-time operation of a charge sensor in a

[1]Center for Quantum Devices, Niels Bohr Institute, University of Copenhagen, 2100 Copenhagen, Denmark. [2]Lorentz Institute and Leiden Institute of Advanced Computer Science, Leiden University, P.O. Box 9506, 2300 RA Leiden, The Netherlands. [3]QDevil, Quantum Machines, 2750 Ballerup, Denmark. [4]Department of Physics and Astronomy, Purdue University, West Lafayette, IN 47907, USA. [5]Birck Nanotechnology Center, Purdue University, West Lafayette, IN 47907, USA. [6]Elmore Family School of Electrical and Computer Engineering, Purdue University, West Lafayette, IN 47907, USA. [7]School of Materials Engineering, Purdue University, West Lafayette, IN 47907, USA. [8]Department of Physics, Norwegian University of Science and Technology, NO-7491 Trondheim, Norway. ✉e-mail: fabrizio.berritta@nbi.ku.dk; anasua.chatterjee@nbi.ku.dk; kuemmeth@nbi.dk

feedback loop[33] maintained the sensor sensitivity for fast charge sensing in a Si/SiGe double quantum dot, compensating for disturbances due to gate-voltage variation and $1/f$ charge fluctuations. A quantum state with higher confidence than what is achievable through traditional thermal methods was initialized by real-time monitoring and negative-result measurements[34].

This study implements real-time two-axis control of a qubit with two fluctuating Hamiltonian parameters that couple to the qubit along different directions on its Bloch sphere. The protocol involves two key steps: first, rapid estimation of the instantaneous magnitude of one of the fluctuating fields (nuclear field gradient) effectively creates one qubit control axis. This control axis is then exploited to probe in real time the qubit frequency (Heisenberg exchange coupling) across different operating points (detuning voltages). Our procedure allows for counteracting fluctuations along both axes, resulting in an improved quality factor of coherent qubit rotations.

Our protocol integrates a singlet-triplet ($ST_0$) spin qubit implemented in a gallium arsenide double quantum dot (DQD)[29] with Bayesian Hamiltonian estimation[35–39]. Specifically, an FPGA-powered quantum orchestration platform (OPX)[40] repeatedly separates singlet-correlated electron pairs using voltage pulses and performs single-shot readout classifications to estimate on-the-fly the fluctuating nuclear field gradient within the double dot[41]. Knowledge of the field gradient in turn enables the OPX to coherently rotate the qubit between S and $T_0$ by arbitrary, user-defined target angles. Differently from previous works, we let the gradient freely fluctuate, without pumping the nuclear field[42], and instead program the OPX to adjust the baseband control pulses accordingly.

An adaptive second-axis estimation is performed to also probe the exchange interaction between the two electrons. This exchange interaction estimation scheme is not simply an independent repetition of the single-axis estimation protocol[35–39]: the design of the exchange-based free induction decay (FID) pulse sequence depends on the outcome of the first-axis estimation and needs to be computed on the fly. Finally, fluctuations along both axes are measured and corrected, enabling the stable coherent rotation of the qubit around a symmetric axis, essential for performing the Hadamard gate.

Our work introduces a versatile method for enhancing coherent control and stability of spin qubits by harnessing low-frequency environmental fluctuations coupling to the system. As such, it is not limited to the operation of $ST_0$ qubits in GaAs. Our implementation of real-time reaction to fluctuating Hamiltonian parameters can find application in other materials and qubit encodings, as it is not necessarily limited to nuclear noise.

## Results

### Device and Bayesian estimation

We use a top-gated GaAs DQD array[29] and tune up one of its $ST_0$ qubits using the gate electrodes shown in Fig. 1c, at 200 mT in-plane magnetic field in a dilution refrigerator with a mixing-chamber plate below 30 mK. Radio-frequency reflectometry off the sensor dot's ohmic contact distinguishes the relevant charge configurations of the DQD[43].

The qubit operates in the (1,1) and (0,2) charge configuration of the DQD. (Integers indicate the number of electrons in the left and right dot.) The electrical detuning $\varepsilon$ quantifies the difference in the electrochemical potentials of the two dots, which in turn sets the qubit's spectrum as shown in Fig. 1a. We do not plot the fully spin-polarized triplet states, which are independent of $\varepsilon$ and detuned in energy by the applied magnetic field. We define $\varepsilon = 0$ at the measurement point close to the interdot (1,1)-(0,2) transition, with negative $\varepsilon$ in the (1,1) region. In the $ST_0$ basis, we model the time-dependent Hamiltonian by

$$\mathcal{H}(t) = J(\varepsilon(t)) \frac{\sigma_z}{2} + g^* \mu_B \Delta B_z(t) \frac{\sigma_x}{2}, \quad (1)$$

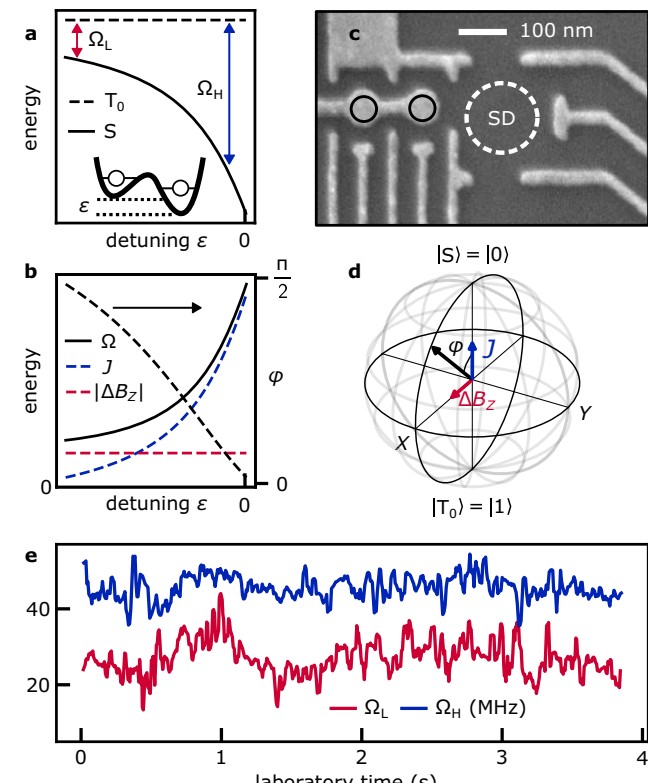

**Fig. 1 | A singlet-triplet ($ST_0$) qubit with two fluctuating control axes. a** The dots' electrical detuning $\varepsilon$ tunes from a regime of low qubit frequency, $\Omega_L$, to a regime of high frequency, $\Omega_H$. States outside the computational space are not plotted. **b** In the first (second) regime, the Overhauser gradient $|\Delta B_z|$ (the exchange coupling $J$) dominates the qubit frequency $\Omega_L$ ($\Omega_H$) and the polar angle $\varphi$ of the qubit rotation axis. **c** SEM image of the GaAs device[29], implementing a two-electron double quantum dot (black circles) next to its sensor dot (SD) for qubit readout. **d** $J$ and $\Delta B_z$ drive rotations of the qubit around two orthogonal axes, providing universal qubit control, as depicted in the Bloch sphere. **e** Uncontrolled fluctuations of the Larmor frequencies $\Omega_L$ and $\Omega_H$, estimated in real time on the OPX and plotted with a 30 ms moving average.

which depends on the detuning $\varepsilon$ that controls the exchange interaction between the two electrons, $J(\varepsilon(t))$, and the component of the Overhauser gradient parallel to the applied magnetic field between the two dots, $\Delta B_z(t)$. $\sigma_i$ are the Pauli operators, $g^*$ is the effective g-factor, and $\mu_B$ is the Bohr magneton. In the following, we drop the time dependence of the Hamiltonian parameters for ease of notation. On the Bloch sphere of the qubit (Fig. 1d), eigenstates of the exchange interaction, $|S\rangle$ and $|T_0\rangle$, are oriented along $Z$, while $\Delta B_z$ enables rotations along $X$.

The qubit is manipulated by voltage pulses applied to the plunger gates of the DQD, and measured near the interdot (1,1)-(0,2) transition by projecting the unknown spin state of (1,1) onto either the (1,1) charge state ($|T_0\rangle$) or the (0,2) charge state ($|S\rangle$). Each single-shot readout of the DQD charge configuration involves generation, demodulation, and thresholding of a few-microsecond-long radio-frequency burst on the OPX (see Supplementary Fig. 1).

The OPX allows for real-time calculation of the qubit Larmor frequency $\Omega(\varepsilon) = \sqrt{\Delta B_z^2 + J(\varepsilon)^2}$ at different detunings, based on real-time estimates of $\Delta B_z$ and $J(\varepsilon)$.

Inspecting the exchange coupling in a simplified Fermi-Hubbard hopping model[23] and inserting $J(\varepsilon)$ into Eq. (1) suggests two physically distinct regimes [Fig. 1b]: At low detuning, in the (1,1) charge state configuration, the Overhauser gradient dominates the qubit dynamics. In this regime, the qubit frequency reads $\Omega_L \equiv \sqrt{\Delta B_z^2 + J_{res}^2}$, where we have added a small phenomenological term $J_{res}$ to account for a constant

residual exchange between the two electrons at low detuning. Such a term may become relevant when precise knowledge of $\Delta B_z$ is required, for example for the Hadamard protocol at the end of this study. At high detuning, close to the (1,1)–(0,2) interdot charge transition, exchange interaction between the two electrons dominates, and the qubit frequency becomes $\Omega_H(\varepsilon) \equiv \sqrt{\Delta B_z^2 + J(\varepsilon)^2}$. As shown in Fig. 1b, the detuning affects both the Larmor frequency $\Omega$ and the polar angle $\varphi$ of the qubit rotation axis $\hat{\boldsymbol{\omega}}$, with $\varphi$ approaching 0 in the limit $J(\varepsilon) \gg \Delta B_z$ and $\pi/2$ if $J(\varepsilon) \ll \Delta B_z$.

Without the possibility of turning off either $J$ or $\Delta B_z$, the rotation axes of the singlet-triplet qubit are tilted, meaning that pure $X$- and $Z$-rotations are unavailable. In their absence, the estimation of the qubit frequency at different operating points is crucial for navigating the whole Bloch sphere of the qubit. Figure 1e tracks Larmor frequencies $\Omega_H$ and $\Omega_L$, both fluctuating over tens of MHz over a period of several seconds, using a real-time protocol as explained later. The presence of low-frequency variations in time traces of $\Omega_H$ and $\Omega_L$ suggests that qubit coherence can be extended by monitoring these uncontrolled fluctuations in real time and appropriately compensating qubit manipulation pulses on-the-fly.

To estimate the frequency of the fluctuating Hamiltonian parameters on the OPX, we employ a Bayesian estimation approach based on a series of free-induction-decay experiments[35]. Using $m_i$ to represent the outcome ($|S\rangle$ or $|T_0\rangle$) of the $i$-th measurement after an evolution time $t_i$, the conditional probability $P(m_i|\Omega)$ is defined as the probability of obtaining $m_i$ given a value of $\Omega$:

$$P(m_i|\Omega) = \frac{1}{2}\left[1 + r_i\left(\alpha + \beta\cos(2\pi\Omega t_i)\right)\right], \qquad (2)$$

where $r_i$ takes a value of 1 ($-1$) if $m_i = |S\rangle$ ($|T_0\rangle$), and $\alpha$ and $\beta$ are determined based on the measurement error and axis of rotation on the Bloch sphere.

Applying Bayes' rule to estimate $\Omega$ based on the observed measurements $m_N, \ldots m_1$, which are assumed to be independent of each other, yields the posterior probability distribution $P(\Omega|m_N, \ldots m_1)$ in terms of a prior uniform distribution $P_0(\Omega)$ and a normalization constant $\mathcal{N}$:

$$\begin{aligned} P(\Omega|m_N, \ldots m_1) = P_0(\Omega)\mathcal{N} \\ \times \prod_{i=1}^{N}\left[1 + r_i\left(\alpha + \beta\cos(2\pi\Omega t_i)\right)\right]. \end{aligned} \qquad (3)$$

Based on previous works[35,38,39], we fix $\alpha = 0.25$ and $\beta = \pm 0.5$, with the latter value positive when estimating $\Omega_L$ and negative when estimating $\Omega_H$. The expectation value $\langle\Omega\rangle$, calculated over the posterior distribution after all $N$ measurements, is then taken as the final estimate of $\Omega$.

## Controlled Overhauser gradient driven rotations

We first implement qubit control using one randomly fluctuating Hamiltonian parameter, through rapid Bayesian estimation of $\Omega_L$ and demonstration of controlled rotations of a $ST_0$ qubit driven by the prevailing Overhauser gradient. Notably, this allows coherent control without a micromagnet[44,45] or nuclear spin pumping[42].

$\Omega_L$ is estimated from the pulse sequence shown in Fig. 2a: for each repetition a singlet pair is initialized in (0,2) and subsequently detuned deep in the (1, 1) region ($\varepsilon_L \approx -40$ mV) for $N = 101$ linearly spaced separation times $t_i$ up to 100 ns. After each separation, the qubit state, $|S\rangle$ or $|T_0\rangle$, is assigned by thresholding the demodulated reflectometry signal $V_{rf}$ near the (1,1)-(0,2) interdot transition and updating the Bayesian probability distribution of $\Omega_L$ according to the outcome of the measurement. After measurement $m_N$, the initially uniform distribution has narrowed [inset of Fig. 2b, with white and black indicating low and high probability], allowing the extraction of

$\langle\Omega_L\rangle$ as the estimate for $\Omega_L$. For illustrative purposes, we plot in Fig. 2a the $N$ single-shot measurements $m_i$ for 10,000 repetitions of this protocol, which span a period of about 20 s, and in Fig. 2b the associated probability distribution $P(\Omega_L)$ of each repetition. The quality of the estimation seems to be lower around a laboratory time of 6 seconds, coinciding with a reduced visibility of the oscillations in panel 2a. We attribute this to an enhanced relaxation of the triplet state during readout due to the relatively high $|\Delta B_z|$ gradient during those repetitions[46]. The visibility could be improved by a latched or shelved read-out[47,48] or energy-selective tunneling-based readout[38].

Even though the rotation speed around $\hat{\boldsymbol{\omega}}_L$ at low detuning is randomly fluctuating in time, knowledge of $\langle\Omega_L\rangle$ allows controlled rotations by user-defined target angles. To show this, we task the OPX in Fig. 2d to adjust the separation times $\tilde{t}_j$ in the pulse sequence to rotate the qubit by $M = 80$ different angles $\theta_j = \tilde{t}_j \langle\Omega_L\rangle$ between 0 and $8\pi$. In our notation, the tilde in a symbol $\tilde{x}$ indicates that the waveform parameter $x$ is computed dynamically on the OPX. To reduce the FPGA memory required for preparing waveforms with nanosecond resolution, we perform controlled rotations only if the expected $\Omega_L$ is larger than an arbitrarily chosen minimum of 50 MHz. (The associated IF statement and waveform compilation then takes about 40 μs on the FPGA.) This reduces the number of precomputed waveforms needed for the execution of pulses with nanosecond-scale granularity, for which we use the OPX baked waveforms capability. Accordingly, the number of rows in Fig. 2d (1450) is smaller than in panel a, and we only label a few selected rows with their repetition number.

To show the increased rotation-angle coherence of controlled $|\Delta B_z|$-driven rotations, we plot the average of all 1450 repetitions of Fig. 2d and compare the associated quality factor, $Q \gtrsim 7$, with that of uncontrolled oscillations, $Q \sim 1$ (we define the quality factor as the number of oscillations until the amplitude is 1/e of its original value). The average of the uncontrolled S-$T_0$ oscillations in Fig. 2a can be fit by a decay with Gaussian envelope (solid line), yielding an inhomogeneous dephasing time $T_2^* \approx 30$ ns typical for $ST_0$ qubits in GaAs[49]. We associate the relatively smaller amplitude of stabilized qubit oscillations with the low-visibility region around 6 seconds in Fig. 2d, discussed earlier. Excluding such regions by post selection increases the visibility and quality factor of oscillations (see Supplementary Fig. 4). Overall, the results presented in this section exemplify how adaptive baseband control pulses can operate a qubit reliably, out of slowly fluctuating environments.

## Real-time two-axis estimation

In addition to nuclear spin noise, $ST_0$ qubits are exposed to electrical noise in their environment, which affects the qubit splitting in particular at higher detunings. It is therefore important to examine and mitigate low-frequency noise at different operating points of the qubit. In the previous section, the qubit frequency $\Omega_L$ was estimated entirely at low detuning where the Overhauser field gradient dominates over the exchange interaction. In order to probe and stabilize also the second control axis, namely $J$-driven rotations corresponding to small $\varphi$ in Fig. 1d, we probe the qubit frequency $\Omega_H$ at higher detunings, using a similar protocol with a modified qubit initialization.

Free evolution of the initial state $|S\rangle$ around $\hat{\boldsymbol{\omega}}_L$ would result in low-visibility exchange-driven oscillations because of the low value of $\varphi$. To circumvent this problem, we precede the $\Omega_H$ estimation by one repetition of $\Omega_L$ estimation, as shown in Fig. 3a. This way, real-time knowledge of $\langle\Omega_L\rangle$ allows the initial state $|S\rangle$ to be rotated to a state near the equator of the Bloch sphere, before it evolves freely for probing $\Omega_H$. This rotation is implemented by a diabatic detuning pulse from (0,2) to $\varepsilon_L$ (diabatic compared to the interdot tunnel coupling) for time $\tilde{t}_{\pi/2}$, corresponding to a rotation of the qubit around $\hat{\boldsymbol{\omega}}_L$ by an angle $\Omega_L \tilde{t}_{\pi/2} = \pi/2$. After evolution for time $t_j$ under finite exchange, another $\pi/2$ rotation around $\hat{\boldsymbol{\omega}}_L$ rotates the qubit to achieve a high readout contrast in the $ST_0$ basis, as illustrated on the Bloch sphere in Fig. 3b.

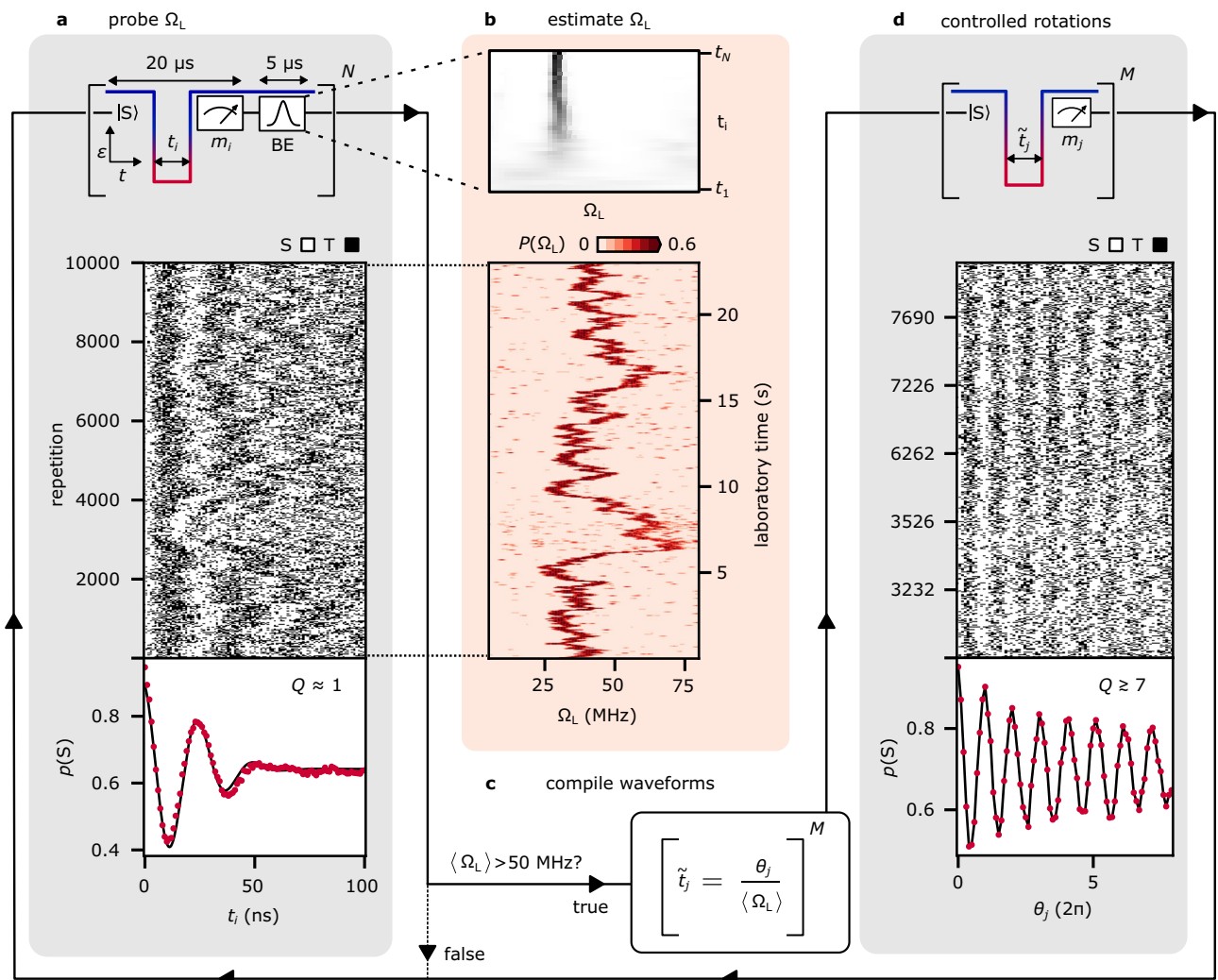

**Fig. 2 | Controlled Overhauser gradient driven rotations of a $ST_0$ qubit by real-time Bayesian estimation.** One loop (solid arrows) represents one repetition of the protocol. **a** For each repetition, the OPX estimates $\Omega_L$ by separating a singlet pair for $N$ linearly spaced probe times $t_i$ and updating the Bayesian estimate (BE) distribution after each measurement, as shown in the inset of b for one representative repetition. For illustrative purposes, each single-shot measurements $m_i$ is plotted as a white/black pixel, here for $N = 101$ $\Omega_L$ probe cycles, and the fraction of singlet outcomes in each column is shown as a red dot. **b** Probability distribution

$P(\Omega_L)$ after completion of each repetition in a. Extraction of the expected value $\langle\Omega_L\rangle$ from each row completes $\Omega_L$ estimation. **c** For each repetition, unless $\langle\Omega_L\rangle$ falls below a user-defined minimum (here 50 MHz), the OPX adjusts the separation times $\tilde{t}_j$, using its real-time knowledge of $\langle\Omega_L\rangle$, to rotate the qubit by user-defined target angles $\theta_j = \tilde{t}_j \langle\Omega_L\rangle$. **d** To illustrate the increased coherence of Overhauser gradient driven rotations, we task the OPX to perform $M = 80$ evenly spaced $\theta_j$ rotations. Single-shot measurements $m_j$ are plotted as white/black pixels, and the fraction of singlet outcomes in each column is shown as a red dot.

As a side note, we mention that in the absence of knowledge of the Overhauser field gradient, the qubit would traditionally be initialized near the equator by adiabatically reducing detuning from (0,2) to the (1,1) charge configuration, and a reverse ramp for readout. Such adiabatic ramps usually last several microseconds each, while our $\tilde{t}_{\pi/2}$ pulses typically take less than 10 ns, thereby significantly shortening each probe cycle.

For the estimate of $\Omega_H$, the Bayesian probability distribution of $\Omega_H$ is updated after each of the $M = 101$ single-shot measurement $m_j$, each corresponding to a separation time $t_j$ that is evenly stepped from 0 to 100 ns. The Bayesian probability distributions of both $\Omega_L$ and $\Omega_H$ are shown in Fig. 3c and d, respectively, with the latter being conditioned on 20 MHz < $\langle\Omega_L\rangle$ < 40 MHz to reduce the required FPGA memory.

This section demonstrated a real-time baseband control protocol that enables manipulation of a spin qubit on the entire Bloch sphere.

### Controlled exchange-driven rotations
Using Bayesian inference to estimate control axes in real-time offers new possibilities for studying and mitigating qubit noise at all

detunings. Figure 4a describes the real-time controlled exchange-driven rotations protocol aimed at stabilizing frequency fluctuations of the qubit at higher detunings. Following the approach of Fig. 3, we first estimate $\Omega_L$ and $\Omega_H$ using real-time Bayesian estimation. We then use our knowledge of $\Omega_H$ to increase the rotation angle coherence of the qubit where the exchange coupling is comparable with the Overhauser field gradient.

As illustrated in Fig. 4a, the qubit control pulses now respond in real time to both qubit frequencies $\Omega_L$ and $\Omega_H$. Similar to the previous section, after determining $\langle\Omega_L\rangle$ and confirming that 30 MHz < $\langle\Omega_L\rangle$ < 50 MHz is fulfilled, the qubit is initialized near the equator of the Bloch sphere by fast diabatic $\Omega_L(\pi/2)$ pulses, followed by an exchange-based FID that probes $\Omega_H$. Based on the resulting $\langle\Omega_H\rangle$, the OPX adjusts the separation times $\tilde{t}_l$ to rotate the qubit by user-defined target angles $\theta_l = \tilde{t}_l \langle\Omega_H\rangle$.

To show the resulting improvement of coherent exchange oscillations, we plot in Fig. 4c the interleaved $K = 101$ measurements $m_l$ and compare them in Fig. 4b to the $M = 101$ measurements $m_j$. Fitting the average of the uncontrolled rotations by an oscillatory fit with

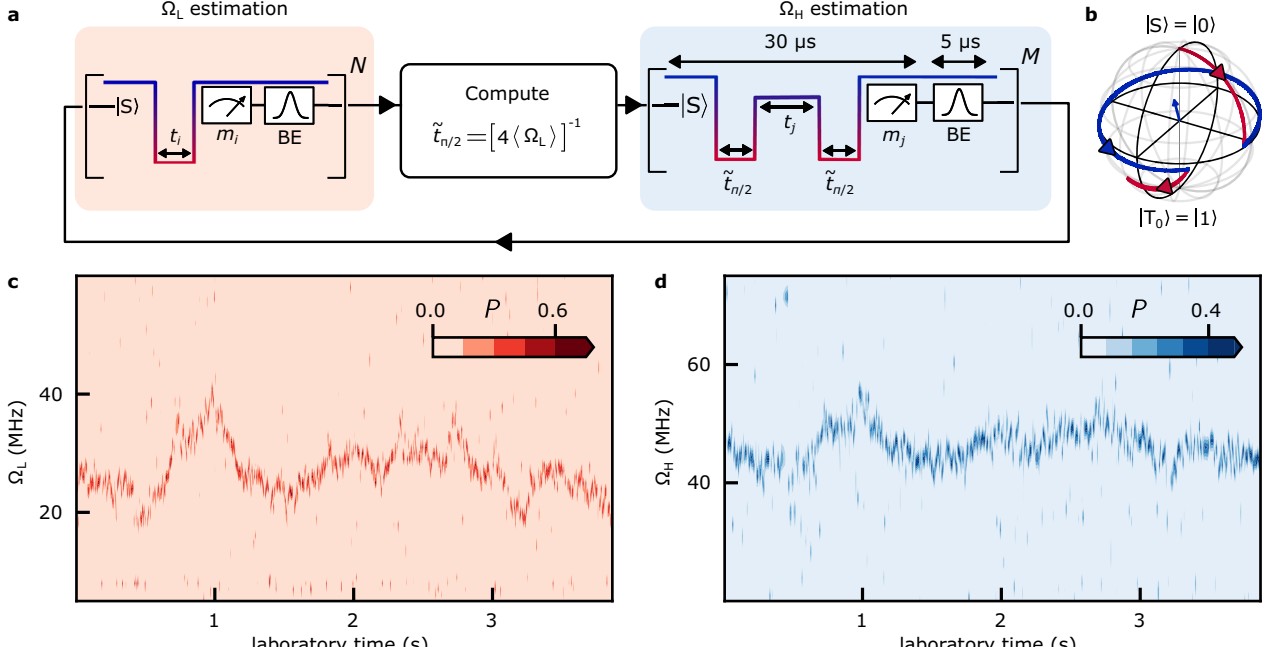

**Fig. 3 | Real-time Bayesian estimation of two control axes. a** One repetition of the two-axis estimation protocol. After estimating $\Omega_L$ from $N = 101$ $t_i$ probe cycles (Fig. 2a), the OPX computes on-the-fly the pulse duration $\tilde{t}_{\pi/2}$ required to initialize the qubit near the equator of the Bloch sphere by a diabatic $\Omega_L(\pi/2)$ pulse. After the $\Omega_L(\pi/2)$ pulse, the qubit evolves for time $t_j$ under exchange interaction before another $\Omega_L(\pi/2)$ pulse initiates readout. After each single-shot measurement $m_j$, the OPX updates the BE distribution of $\Omega_H$. Similar to $t_i$ in the $\Omega_L$ estimation, $t_j$ is spaced evenly between 0 and 100 ns across $M = 101$ exchange probe cycles. **b** Qubit evolution on the Bloch sphere during one exchange probe cycle. **c** Each column plots $P(\Omega_L)$ after completion of the $\Omega_L$ estimation in each protocol repetition. **d** Each column plots $P(\Omega_H)$ after completion of the $\Omega_H$ estimation in each protocol repetition.

Gaussian envelope decay yields $T_{el}^* \approx 60$ ns and $Q \approx 3$, presumably limited by electrical noise[49], while the quality factor of the controlled rotations is enhanced by a factor of two, $Q \approx 6$.

The online control of exchange-driven rotations using Bayesian inference stabilizes fluctuations of the qubit frequency at higher detunings, where fluctuations are more sensitive to detuning noise. Indeed, we attribute the slightly smaller quality factor, relative to Overhauser-driven rotations in Fig. 2d, to an increased sensitivity to charge noise at larger detuning, which, owing to its high-frequency component, is more likely to fluctuate on the estimation timescales[36].

This section established for the first time stabilization of two rotation axes of a spin qubit. This advancement should allow for stabilized control over the entire Bloch sphere, which we demonstrate in the next section.

**Hadamard rotations**

In this experiment, we demonstrate universal $ST_0$ control that corrects for fluctuations in all Hamiltonian parameters. We execute controlled Hadamard rotations around $\hat{\omega}_{Had}$, as depicted by the trajectory on the Bloch sphere of Fig. 5d, by selecting the detuning $\varepsilon_{Had}$ in real time such that $J(\varepsilon_{Had}) = |\Delta B_z|$. To achieve this, we do not assume that $\Delta B_z = \Omega_L$ (i.e. we allow contributions of $J_{res}$ to $\Omega_L$) or that $J = \Omega_L$ (i.e. we allow contributions of $\Delta B_z$ to $\Omega_H$). The full protocol is detailed in Supplementary Discussion.

Real-time knowledge of both $\Delta B_z$ and $J$ would potentially benefit two-qubit gate fidelities[50] and the resonant-driving approach of previous works[35,38,39]. In the resonant implementation, constrained to the operating regime $|\Delta B_z| \gg J$, low-frequency fluctuations of $J$ result in transverse noise that causes dephasing and phase shifts of the Rabi rotations[51,52].

In previous sections, we have shown how to probe the qubit Larmor frequencies $\Omega_H$ and $\Omega_L$ at different detunings in real time and correct for their fluctuations. Now, we simultaneously counteract fluctuations in $J$ and $|\Delta B_z|$ on the OPX in order to perform the

Hadamard gate. As we do not measure the sign of $\Delta B_z$, we identify the polar angle of $\hat{\omega}_{Had}$ as either $\varphi = \pi/4$ or $-\pi/4$. In other words, starting from the singlet state, the qubit rotates towards $+X$ on the Bloch sphere for one sign of $\Delta B_z$, and towards $-X$ for the other sign. The sign of the gradient may change over long time scales due to nuclear spin diffusion (on the order of many seconds[41]), but the measurement outcomes of our protocol are expected to be independent of the sign.

The relative sign of Overhauser gradients becomes relevant for multi-qubit experiments[53], and could be determined[54] by comparing the relaxation time of the ground state (e.g. $|\uparrow\downarrow\rangle$) of $\Delta B_z$ with its excited state ($|\downarrow\uparrow\rangle$). Such diagnostic sign-probing cycles on the FPGA should not require more than a few milliseconds, negligible compared to the expected time between sign reversals.

In preparation for our protocol, we first extract the time-averaged exchange profile by performing exchange oscillations as a function of evolution time [Fig. 5b]. Removing contributions of $\Delta B_z$ to $\Omega_H$ then yields $J(\varepsilon)$ in Fig. 5c. A linear approximation in the target range 40 MHz $< \langle J \rangle <$ 60 MHz (dashed blue line) is needed later on the OPX to allow initial detuning guesses when tuning up $J(\varepsilon) = |\Delta B_z|$. We also provide the OPX with a value for the residual exchange at low detuning, $J_{res} \approx 20$ MHz, determined offline as described in Supplementary Fig. 5.

As illustrated in Fig. 5a, the Hadamard rotation protocol starts by estimating $|\Delta B_z|$ from $\Omega_L$, taking into account a constant residual exchange by solving $\Delta B_z^2 = \Omega_L^2 - J_{res}^2$.

Next, an initial value of $\varepsilon_{Had}$ is chosen based on the linear offline model to fulfill $J(\varepsilon_{Had}) = |\Delta B_z|$ [feedback in panel (a,c)]. To detect any deviations of the prevailing $J$ from the offline model, an exchange-driven FID is performed at $\varepsilon_{Had}$ to estimate $J$ from $\Omega_H$, using $J^2 = \Omega_H^2 - \Delta B_z^2$.

Any deviation of $\langle J \rangle$ from the target value $|\Delta B_z|$ is subsequently corrected for by updating $\varepsilon_{Had}$ based on the linearized $J(\varepsilon)$ model [feedback in panels (a,c)]. Matching $J$ to $|\Delta B_z|$ in the two detuning feedback steps each takes about 400 ns on the OPX. Finally, real-time

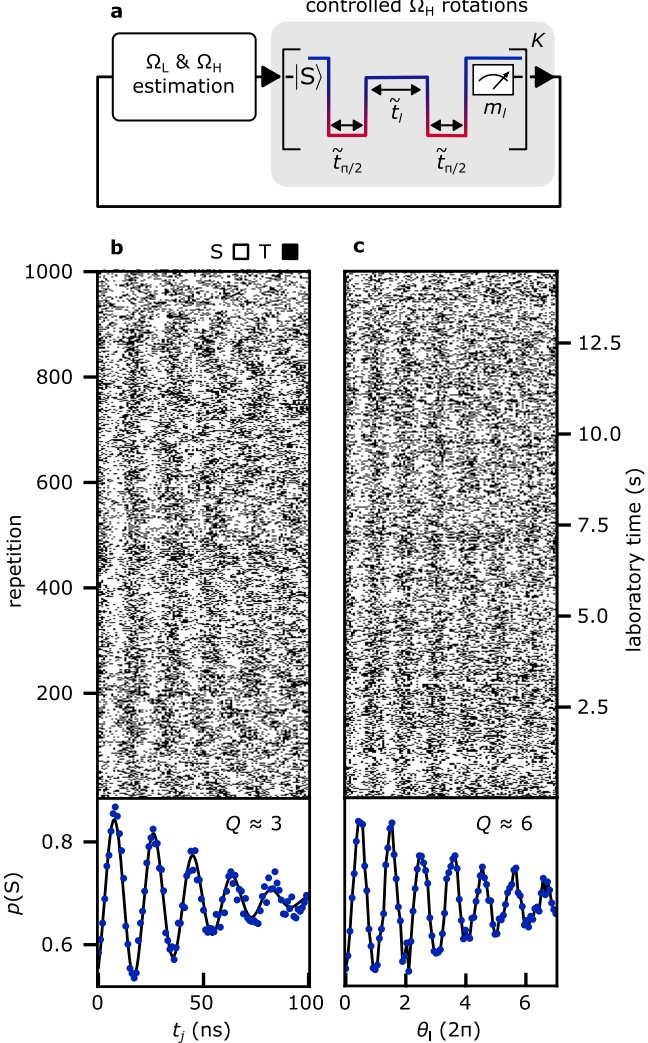

**Fig. 4 | Real-time-controlled exchange-driven qubit rotations. a** One repetition of the exchange rotation protocol. After estimation of $\Omega_L$ and $\Omega_H$ as in Fig. 3, the OPX adjusts exchange duration times $\tilde{t}_l$, using real-time knowledge of $\langle\Omega_H\rangle$, to rotate the qubit by user-defined target angles $\theta_l = \tilde{t}_l \langle\Omega_H\rangle$. Pulse durations $\tilde{t}_{\pi/2}$ for qubit initialization and readout use real-time knowledge of $\langle\Omega_L\rangle$. **b** Each row plots measurements $m_j$ from one protocol repetition, here $M = 101$ exchange probe outcomes. **c** Each row plots measurement $m_l$ from one protocol repetition, here $K = 101$ controlled-exchange-rotation outcomes. To illustrate the increased coherence of controlled exchange rotations, we also plot in b and c the fraction of singlet outcomes of each column.

knowledge of $\Omega_{Had} \equiv \sqrt{2}|\Delta B_z|$ is used to generate the free evolution times $\tilde{t}_i$, spent at the updated value $\tilde{\varepsilon}_{Had}$, in order to perform Hadamard rotations by $K$ user defined target angles.

The resulting Hadamard oscillations are shown in Fig. 5e (top panel) and fitted with an exponentially decaying sinusoid, indicating a quality factor $Q > 5$. (According to this naive fit, the amplitude drops to $1/e$ over approximately 40 rotations, although we have not experimentally explored rotation angles beyond $9\pi$.) In comparison to exchange-controlled rotations from Fig. 4c, the Hadamard rotations are more stable, which we attribute to the additional feedback on detuning that fixes the oscillation axis and decreases sensitivity to charge noise.

To illustrate the crucial role of real-time estimation for this experiment, we also performed rotation experiments that do not involve any real-time estimation and feedback cycles (dark gray data, bottom panel), as follows. Within minutes after performing the controlled Hadamard rotations (purple data), we executed Hadamard

rotations assuming a fixed value of $|\Delta B_z| = \overline{|\Delta B_z|}$, i.e. by pulsing to a fixed detuning value corresponding to $J(\varepsilon_H) = \overline{|\Delta B_z|}$ according to the offline model. Here, $\overline{|\Delta B_z|} \approx 40$ MHz is the average Overhauser gradient that we observed just before executing the Hadamard protocol. Not surprisingly, the quality factor of the resulting Hadamard-like oscillations is low and the rotation angle deviates from the intended target angle, likely due to the Overhauser gradient having drifted in time. As a side note, we mention that the purple data in Fig. 5e constitutes an average over 5000 repetitions, corresponding to a total acquisition time of 2 minutes including Overhauser and exchange estimation cycles. In contrast, the dark gray data also constitutes an average over 5000 repetitions, but only required 15 seconds because of the omission of all estimation and feedback cycles.

To verify that the enhancement in $Q$ is not solely due to the more accurate knowledge of $|\Delta B_z|$, we also performed Hadamard rotations only using the estimation of $|\Delta B_z|$. The FPGA was programmed to perform a measurement where the initialized singlet is pulsed to a fixed detuning $J(\varepsilon_H) \approx 20$ MHz to perform a Hadamard rotation, only if the estimated $|\Delta B_z|$ on the FPGA satisfies $17\,\mathrm{MHz} < \langle|\Delta B_z|\rangle < 23\,\mathrm{MHz}$. We then post select the repetitions where $19.5\,\mathrm{MHz} < \langle|\Delta B_z|\rangle < 20.5\,\mathrm{MHz}$. Fitting this by an oscillatory fit with Gaussian envelope decay yields $T_{el}^* \approx 70$ ns, $Q \approx 2.0$ and frequency $\approx 29$ MHz.

In Fig. 5e we compare these data (light gray, middle panel) with the cases where the FPGA estimated both $|\Delta B_z|$ and $J$ (purple, top panel) and where the microprocessor does not perform any estimation but simply pulses to $J(\varepsilon_H)$ to perform the rotations (dark gray, bottom panel). (In the middle panel the horizontal axis was rescaled to the Hadamard evolution time using the fitted frequency $\approx 29$ MHz.) We see that (i) a reduction of the uncertainty in $|\Delta B_z|$ from $\approx 30$ MHz (r.m.s.) to $\approx 2$ MHz (dark gray to light gray) does not yield a proportional gain in $Q$ and (ii) the improvement in $Q$ when including estimation of $J$ (light gray to purple) is much larger than can be justified solely by the slight further reduction of the uncertainty in $|\Delta B_z|$ (roughly from $\approx 2$ MHz to $\approx 1$ MHz). This demonstrates the crucial contribution of the estimations along both axes in the improvement of our Hadamard gate quality factor.

Further evidence for the fluctuating nature of non-stabilized Hadamard rotations is discussed in Supplementary Fig. 6.

The stabilized Hadamard rotations demonstrate real-time feedback control based on Bayesian estimation of $J$ and $|\Delta B_z|$, and suggest a significant improvement in coherence for $ST_0$ qubit rotations around a tilted control axis. Despite the presence of fluctuations in all Hamiltonian parameters, we report effectively constant amplitude of Hadamard oscillations, with a reduced visibility that we tentatively attribute to estimation and readout errors.

## Discussion

Our experiments demonstrate the effectiveness of feedback control in stabilizing and improving the performance of a singlet-triplet spin qubit. The protocols presented showcase two-axis control of a qubit with two fluctuating Hamiltonian parameters, made possible by implementing online Bayesian estimation and feedback on a low-latency FPGA-powered qubit control system. Real-time estimation allows control pulses to counteract fluctuations in the Overhauser gradient, enabling controlled Overhauser-driven rotations without the need for micromagnets or nuclear polarization protocols. Notably, even in the absence of a deterministic component of the Hamiltonian purely noise-driven coherent rotations of a two-level quantum system were demonstrated.

The approach is extended to the real-time estimation of the second rotation axis, dominated by exchange interaction, which we then combine with an adaptive feedback loop to generate and stabilize Hadamard rotations. In particular, executing the Hadamard gate involves (i) sequentially executing two distinct estimation cycles, where the design of the second cycle relies on the outcomes of the

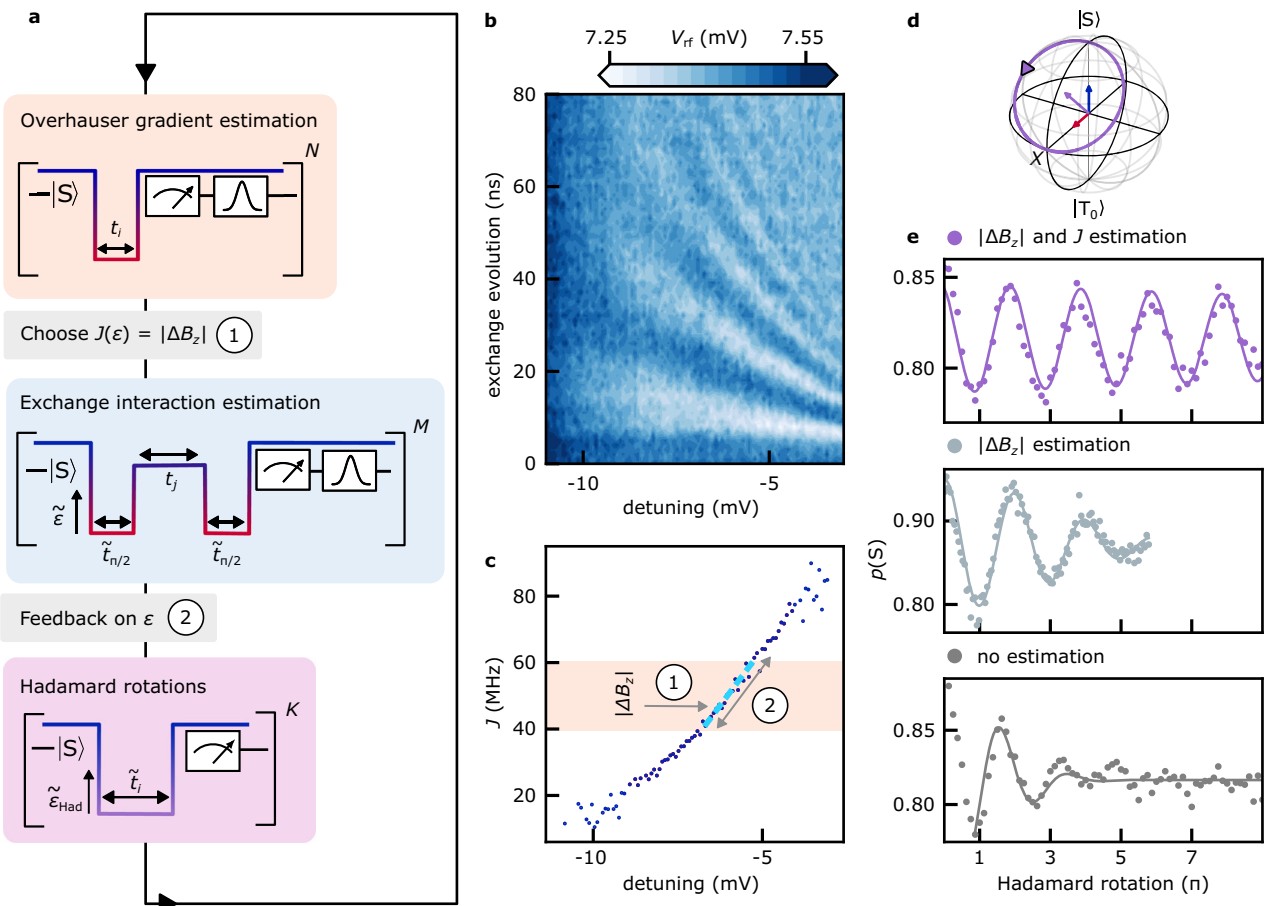

**Fig. 5 | Real-time universal $ST_0$ control demonstrated by Hadamard rotations.**
**a** Hadamard rotation protocol. After estimating $\Omega_L$, $\varepsilon$ is chosen in real-time such that $J(\varepsilon) = |\Delta B_z|$, based on a linearized offline model from panel c. If $40\,\text{MHz} < \langle |\Delta B_z| \rangle < 60\,\text{MHz}$, the detuning is adjusted to account for deviations of the prevailing $J$ from the offline model. Real-time knowledge of $\Omega_{\text{Had}} = \sqrt{2}\,|\Delta B_z|$ then dictates $\tilde{t}_i$ to achieve a user-defined Hadamard rotation angle. **b** Averaged exchange driven FID as a function of detuning and evolution time. Here, a diabatic

$\Omega_L(\pi/2)$ pulse initializes the qubit near the equator of the Bloch sphere, prior to free exchange evolution, and subsequently prepares it for readout. **c** $J$ as a function of $\varepsilon$ extracted offline from b, as well as a linearized model (dashed line) used in the two feedback steps of panel a. **d** Hadamard rotation depicted on the Bloch sphere. **e** Measurement of Hadamard rotations with $|\Delta B_z|$ and $J$ estimation (purple, top panel), only $|\Delta B_z|$ estimation (light gray, middle panel), and without the feedback shown in a (dark gray, bottom panel).

first, (ii) correlating the detected frequencies to distinguish independent fluctuations of the two control axes, and (iii) utilizing this correlated information to dynamically construct and execute a Hadamard gate. These steps demand real-time adaptive estimations and signal generations throughout the protocol, which has not been demonstrated before. A constant Overhauser field gradient, whether stemming from nuclear spin pumping or a micromagnet, is expected to further improve the feedback control. From this perspective, our work represents a worst-case scenario, demonstrating the effectiveness of our experimental technique.

Our protocols assume that $\Delta B_z$ does not depend on the precise dot detuning in the (1,1) configuration and remains constant on the time scale of one estimation. Similarly, stabilization of exchange rotations is only effective for electrical fluctuations that are slow compared to one estimation. Therefore, we expect potential for further improvements by more efficient estimation methods, for example through adaptive schemes[55] for Bayesian estimation from fewer samples, or by taking into account the statistical properties of a time-varying signal described by a Wiener process[56] or a nuclear spin bath[57]. Machine learning could be used to predict the qubit dynamics[58–60], possibly via long short-term memory artificial neural networks as reported for superconducting qubits[61]. While our current qubit cycle time (approximately 30 µs) is dominated by readout and qubit initialization, it can potentially be reduced to a few

microseconds through faster qubit state classification, such as enhanced latched readout[48], and faster reset, such as fast exchange of one electron with the reservoir[62]. Our protocol could be modified for real-time non-local noise correlations[63] or in-situ qubit tomography using fast Bayesian tomography[64] to study the underlying physics of the noisy environment, thereby providing qualitatively new insights into processes affecting qubit coherence and multi-qubit error correction.

Beyond $ST_0$ qubits, our protocols uncover new perspectives on coherent control of quantum systems manipulated by baseband pulses. This work represents a significant advancement in quantum control by implementing an FPGA-powered technique to stabilize in real time the qubit frequency at different manipulation points.

## Methods
### Experimental setup
We use an Oxford Instruments Triton 200 cryofree dilution refrigerator with base temperature below 30 mK. The experimental setup employs a Quantum Machines OPX+ for radio-frequency (RF) reflectometry and gate control pulses. The RF carrier frequency is ≈158 MHz and the gate control pulses sent to the left and right plunger gates of the DQD are filtered with low-pass filters (≈220 MHz) at room temperature, before being attenuated at different stages of the refrigerator. Low-frequency tuning voltages (high-frequency baseband

waveforms) are applied by a QDAC[65] (OPX) via a QBoard high-bandwidth sample holder[66].

## Measurement details

Before qubit manipulation, an additional reflectometry measurement is taken as a reference to counteract slow drifts in the sensor dot signal. At the end of each qubit cycle, a $\approx 1\,\mu s$ long pulse is applied to discharge the bias tee. As the qubit cycle period (tens of $\mu s$) is much shorter than the bias tee cutoff ($\approx 300\,Hz$), we do not correct the pulses for the transfer function of the bias tee.

## Data availability

The datasets generated and analyzed during the current study are available from the corresponding authors (F.B., A.C., and F.K.) upon request.

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

## Acknowledgements

This work received funding from the European Union's Horizon 2020 research and innovation programme under grant agreements 101017733 (QuantERA II) and 951852 (QLSI), from the Novo Nordisk Foundation under Challenge Programme NNF20OC0060019 (SolidQ), from the Inge Lehmann Programme of the Independent Research Fund Denmark, from the Research Council of Norway (RCN) under INTFELLES-Project No 333990, as well as from the Dutch National Growth Fund (NGF) as part of the Quantum Delta NL programme.

## Author contributions

F.B. lead the measurements and data analysis, and wrote the manuscript with input from all authors. F.B., T.R., J.v.d.H., A.C. and F.K. performed the experiment with theoretical contributions from J.A.K., J.D. and E.v.N. F.F. fabricated the device. S.F., G.C.G. and M.J.M. supplied the heterostructures. A.C. and F.K. supervised the project.

## Competing interests

The authors declare no competing interests.
