## [Peer Review File · Nature Communications]

REVIEWER COMMENTS

Reviewer #1 (Remarks to the Author):

In this work, F. Berritta et al. demonstrate an FPGA-based feedback-based adaptive control of a GaAs singlet-triplet qubit. Central to the experimental technique, the feedback is based on fast frequency estimation by a series of Bayesian inference steps. Using this, they showed improvement of quantum oscillation quality by a factor of a few compared with non-adaptive control where the exact improvement factor varies depending on the control axis on the Bloch sphere.

On the positive side, I am convinced that the methodology used in the work is solid and the manuscript is logical. It also adequately identifies relevant previous works. However, overall, I argue that the work's novelty and significance are insufficient for publication in Nature Communications. My opinion is based on the following reasons.

1. Novelty: As the authors properly cited, there are plenty of examples before the current work that successfully demonstrated Bayesian-based feedback control of GaAs spin qubits (both single-spin and singlet-triplet type). The claimed new element, estimating δB and J in the same qubit manipulation run, in my opinion, is incremental progress over the previous works. While I believe a Bayesian inference-based FPGA circuit is useful and will be useful for the efficient calibration of some parts of quantum circuits, the methodology and experimental technique are another adaptation of a well-known method to a well-known GaAs qubit.

2. Estimating J , efficient? : Related to #1, I am not convinced whether the estimation of J (after estimating δB) leads to a significant improvement in quantum control quality. Ref. 38 already showed that just estimating δB fast and accurately leads to J -driven oscillations with $Q > 20$ even without estimating J by Bayesian circuit. As Fig 5e only compares estimation-on and completely-off cases (I assume in this case both δB and J estimations are off), it is possible that improvement of Q just stems from narrowing δB distribution, in which case, J estimation actually does not play a role.

3. Base-band two-axis control, necessary?: While the sequential application of two Bayesian estimations and adaptive control of time duration (pulse width) is a different approach to enable two-axis control from previous relevant works, I wonder if this scheme has significantly bigger merit over other methods such as ac-driven two-axis control where only δB estimation is typically needed and just rf phase control is enough to enable two-axis control. In fact, the improvements of Q shown in Refs#35,38, and 39 are better than what is demonstrated in this work. The current work does not use micromagnet or nuclear pumping, but I do not see a significant advantage in this. Because of the

absence of such techniques, dB can be very small occasionally, and dB in this work needed to be conditioned (or heralded): waiting time is presumably very large.

4. Is this a scalable approach?: I can easily imagine excessive experimental overhead when one attempts to do a multi-qubit experiment with this technique. Not only very big FPGA resources are needed for just a few-qubit control, but also many independent sensors are required since the technique fundamentally relies on fast projective measurement on each qubit. The latter is also a fundamental bottleneck of the scheme: As this is measurement-based feedback control, it is classical feedback, or efficient calibration problem, not generally adaptable while performing coherent operations. Moreover, exponentially increasing waiting time is expected due to the need for simultaneous heralding of meeting the right conditions for all qubit frequencies.

5. Using commercially available FPGA, significant improvement in estimation speed?: Overall, I find that the speed of estimation, single-shot readout duration, Bayesian calculation time, and pulse controllability (like granularity) are all comparable to existing experiments using home-built FPGA boards. I do not see significant technical improvement on the electronics side.

Reviewer #2 (Remarks to the Author):

The authors reported,

- i. Real-time two-axis control of a GaAs S-T0 qubit utilizing FPGA fast feedback control
- ii. Controlled Overhauser gradient driven rotations
- iii. Controlled exchange driven rotations
- iv. Demonstration of Hadamard rotations utilizing the two-axis control

I think that these are novel and of interest to readers in the community.

However, the following issues should be considered because these might relate to the main point of this paper.

Measurement setup

1. Indicating the position of $\epsilon = 0$ in Fig. 1a and b will help readers to understand the situation.

Controlled Overhouser gradient driven rotations and related operations

2. As the author mentioned in the Hadamard section, the sign of ΔB_z will not be obtained in this scheme. What is the typical timescale of the change of the sign?

3. For future complicated gate operations, readers would like to know whether the lack of information on the sign of ΔB_z affects the operations. Can the authors comment on a good method to obtain the sign? Is the method presented in Ref. 50 enough for practical applications in terms of the required time, for example?

4. For future complicated gate operations (not only for the Hadamard gate), can the authors comment on a scheme that does not depend on the sign of ΔB_z or is robust against the change of the sign?

Hadamard rotations

5. The improvement of the Q factor ($Q > 5$, naively $Q \sim 40$) in Hadamard rotations looks much better than those in the cases of Overhouser gradient driven rotations ($Q \geq 7$) and exchange driven rotations ($Q \sim 6$). What is the reason of this better performance especially in the case of the Hadamard rotations compared to the other cases?

Reviewer #3 (Remarks to the Author):

The manuscript entitled "Real-time two-axis control of a spin qubit" by Berritt et al. discusses the implementation of real-time two-axis control of a qubit with two fluctuating Hamiltonian parameters: the qubit frequency and the J-driven rotations. The proposed method allows for counteracting fluctuations along both axes, resulting in an improved quality factor of coherent qubit rotations. They show that adaptive baseband control pulses can reliably operate a qubit out of slowly fluctuating environments. They also show how they can examine and mitigate low-frequency noise at different operating points of the qubit. The effectiveness of the proposed protocol to stabilize and improve the performance of a singles-triples qubit is demonstrated in a series of nice experiments, from the controlled rotation of the ST_0 qubit to real-time universal ST_0 control, with the qubit being stabilized in real time using FPGA-powered techniques.

Overall the paper is very well written and presents an important contribution to the field of the control of spin-based qubits. I believe it is well suited for publication in Nature Communications.

We thank the reviewers for their careful consideration of the manuscript. Below we list our point-by-point responses to all comments from the Reviewers. All changes in the manuscript are highlighted in red. We also note that Figure 5e is modified with additional data. We believe that all the comments in the Reviewers’ reports are addressed due to this extensive revision and that our manuscript is now appropriate for publication in *Nature Communications*.

Response to Reviewer #1:

Comment #0

In this work, F. Berritta et al. demonstrate an FPGA-based feedback-based adaptive control of a GaAs singlet-triplet qubit. Central to the experimental technique, the feedback is based on fast frequency estimation by a series of Bayesian inference steps. Using this, they showed improvement of quantum oscillation quality by a factor of a few compared with non-adaptive control where the exact improvement factor varies depending on the control axis on the Bloch sphere.

On the positive side, I am convinced that the methodology used in the work is solid and the manuscript is logical. It also adequately identifies relevant previous works. However, overall, I argue that the work’s novelty and significance are insufficient for publication in Nature Communications. My opinion is based on the following reasons.

Response: We thank the reviewer for carefully reading the manuscript and finding it solid and logical. We have modified the manuscript to address all the comments raised by the reviewer.

Comment #1

1. Novelty: As the authors properly cited, there are plenty of examples before the current work that successfully demonstrated Bayesian-based feedback control of GaAs spin qubits (both single-spin and singlet-triplet type). The claimed new element, estimating dB and J in the same qubit manipulation run, in my opinion, is incremental progress over the previous works. While I believe a Bayesian inference-based FPGA circuit is useful and will be useful for the efficient calibration of some parts of quantum circuits, the methodology and experimental technique are another adaptation of a well-known method to a well-known GaAs qubit.

Response:

We believe performing two-axis estimation is not simply incremental progress compared to single-axis estimation: instantaneous knowledge about two independent control axes allows in principle for full and reliable control over a qubit coupled to a low-frequency noisy environment, whereas single-axis estimation can only be used to perform one specific rotation on the Bloch sphere more accurately. Demonstrating single-parameter estimation [Refs. 35-39] was thus merely a proof-of-principle of one of the ingredients needed for full control in the presence of (nuclear) noise, but it left the latter open as a long-term challenge, which has only now been fulfilled. We thus believe that this first spin-qubit experiment with successful two-axis estimation and subsequent improved qubit control presents a milestone of this branch of research rather than an incremental step. We agree with the reviewer that this kind of in-situ efficient calibration will be crucial for the future operation of quantum processors.

We crucially emphasize that combining everything together involves much more than just performing two repetitions of the well-known estimation scheme. To perform the Hadamard gate, the parameters used in consecutive steps of the estimation have to be chosen adaptively based on the previous steps, by:

- performing sequentially two qualitatively different estimation cycles, where the design of the second cycle depends on the outcome of the first;
- correlating the two frequencies detected to disentangle the independent fluctuations of the two control axes;
- using this information to adaptively construct and perform a Hadamard gate.

All these steps have to be performed adaptively on-the-fly, thus constituting a much more complex experiment than *repetitive* and *independent* single-axis-estimation proof-of-principle. This adaptive character of real-time control is qualitatively new and was absent in previous works.

Apart from its specific relevance for ST qubits, we believe the paper brings new insight on coherent control of a general quantum system coherently controlled by baseband pulses. In the absence of a deterministic component of the Hamiltonian, we demonstrate purely *noise-driven coherent rotations* of a two-level quantum system (without narrowing the noise source by, e.g., nuclear pumping). This shows that with efficient estimation and feedback loops, low-frequency noise (typical for solid-state qubits) can be turned into a resource for control qubits or large quantum systems that are difficult to control otherwise. To clarify the advantage, we have modified the manuscript as follows:

Abstract:

Based on the estimated field gradient, it also estimates the exchange interaction between the two electrons and adjusts their detuning, resulting in extended coherence of Hadamard rotations when correcting for fluctuations of both qubit axes.

Our study emphasizes the critical role of feedback in enhancing the performance and stability of quantum devices affected by quasistatic noise, irrespective of whether it originates from nuclear sources or not.

Introduction:

For the first time in spin qubits an adaptive second-axis estimation is performed to also probe the exchange interaction between the two electrons. This exchange interaction estimation scheme is not simply an independent repetition of the single-axis estimation protocol [35-39]: the design of the exchange-based free induction decay (FID) pulse sequence depends on the outcome of the first-axis estimation and needs to be computed on the fly. Finally, fluctuations along both axes are measured and corrected, enabling the stable coherent rotation of the qubit around a symmetric axis, essential for performing the Hadamard gate.

Our work introduces a versatile method for enhancing coherent control and stability of spin qubits by harnessing low-frequency environmental fluctuations coupling to the system. As such, it is not limited to the operation of ST₀ qubits in GaAs. Our implementation of real-time reaction to fluctuating Hamiltonian parameters can find application in other materials and qubit encodings, as it is not necessarily limited to nuclear noise.

Discussion:

Real-time estimation allows control pulses to counteract fluctuations in the Overhauser gradient, enabling controlled Overhauser-driven rotations without the need for micromagnets or nuclear polarization protocols. **Notably, even in the absence of a deterministic component of the Hamiltonian purely noise-driven coherent rotations of a two-level quantum system were demonstrated.**

The approach is extended to the real-time estimation of the second rotation axis, dominated by exchange interaction, which we then combine with an adaptive feedback loop to generate and stabilize Hadamard rotations. **In particular, executing the Hadamard gate involves (i) sequentially executing two distinct estimation cycles, where the design of the second cycle relies on the outcomes of the first, (ii) correlating the detected frequencies to distinguish independent fluctuations of the two control axes, and (iii) utilizing this correlated information to dynamically construct and execute a Hadamard gate. These steps demand real-time adaptive estimations and signal generations throughout the protocol, which has not been demonstrated before.**

[...]

Beyond ST₀ qubits, our protocols uncover new perspectives on coherent control of quantum systems manipulated by baseband pulses. This work represents a significant advancement in quantum control by implementing an FPGA-powered technique to stabilize in real time the qubit frequency at different manipulation points.

Comment #2

2. Estimating J, efficient? : Related to #1, I am not convinced whether the estimation of J (after estimating dB) leads to a significant improvement in quantum control quality. Ref. 38 already showed that just estimating dB fast and accurately leads to J-driven oscillations with $Q > 20$ even without estimating J by Bayesian circuit. As Fig 5e only compares estimation-on and completely-off cases (I assume in this case both dB and J estimations are off), it is possible that improvement of Q just stems from narrowing dB distribution, in which case, J estimation actually does not play a role.

Response: As we explained in response to Comment #1, the main scope of this work is our demonstration of frequency estimation along two independent axes, the on-the-fly disentangling of that information, and its translation into qubit control along a third axis (Hadamard rotation axis, but it could be any other desired axis). Our experiment is a milestone in that sense: we are the first to successfully integrate these steps and then show that they indeed lead to improved full qubit control.

To verify whether the improvement of Q just stems from narrowing dB distribution, we have performed an additional measurement, added the resulting data to Fig. 5e and discussed this in the main text. We show that by narrowing ΔB_z within 3 MHz bandwidth (without J estimation), the obtained quality factor $Q \sim 2$ of the Hadamard rotations is lower than the (stable) rotations shown in Fig.5e ($Q > 5$). We believe this makes the manuscript stronger and thank the reviewer for prompting us to include this. We have modified the manuscript as follows to show this:

Hadamard rotations

[...]

e Measurement of Hadamard rotations with $|\Delta B_z|$ and J estimation (purple, top panel), only $|\Delta B_z|$ estimation (light gray, middle panel), and without the feedback shown in a (dark gray, bottom panel).

[...]

To verify that the enhancement in Q is not solely due to the more accurate knowledge of $|\Delta B_z|$, we also performed Hadamard rotations only using the estimation of $|\Delta B_z|$. The FPGA was programmed to perform a measurement where the initialized singlet is pulsed to a fixed detuning $J_{\epsilon} H \sim 20$ MHz to perform a Hadamard rotation, only if the estimated $|\Delta B_z|$ on the FPGA satisfies $17 \text{ MHz} < |\Delta B_z| < 23 \text{ MHz}$.

We then post select the repetitions where $19.5 \text{ MHz} < |\Delta B_z| < 20.5 \text{ MHz}$. Fitting this by an oscillatory fit with Gaussian envelope decay yields $T_{el} \sim 70$ ns, $Q \sim 2.0$ and frequency ~ 29 MHz.

In Fig. 5e we compare these data (light gray, middle panel) with the cases where the FPGA estimated both $|\Delta B_z|$ and J (purple, top panel) and where the microprocessor does not perform any estimation but simply pulses to $J_{\epsilon} H$ to perform the rotations (dark gray, bottom panel). (In the middle panel the horizontal axis was rescaled to the Hadamard evolution time using the fitted frequency ~ 29 MHz.)

We see that (i) a reduction of the uncertainty in $|\Delta B_z|$ from ~ 30 MHz (r.m.s.) to ~ 2 MHz (dark gray to light gray) does not yield a proportional gain in Q and (ii) the improvement in Q when including estimation of J (light gray to purple) is much larger than can be justified solely by the slight further reduction of the uncertainty in $|\Delta B_z|$ (roughly from 2 MHz to 1 MHz).

This demonstrates the crucial contribution of the estimations along both axes in the improvement of our Hadamard gate quality factor.

Comment #3

3. Base-band two-axis control, necessary?: While the sequential application of two Bayesian estimations and adaptive control of time duration (pulse width) is a different approach to enable two-axis control from previous relevant works, I wonder if this scheme has significantly bigger merit over other methods such as ac-driven two-axis control where only dB estimation is typically needed and just rf phase control is enough to enable two-axis control. In fact, the improvements of Q shown in Refs#35,38, and 39 are better than what is demonstrated in this work. The current work does not use micromagnet or nuclear pumping, but I do not see a significant advantage in this. Because of the absence of such techniques, dB can be very small occasionally, and dB in this work needed to be conditioned (or heralded): waiting time is presumably very large.

Response:

Our work is not meant to promote ST qubits in this sense. It presents a general technique of how to (i) work with spin qubits coupled to a slowly fluctuating environment, (ii) improve full coherent qubit control by adaptive baseband pulses, based on instantaneous knowledge about the environmental fluctuations coupling to σ_x and σ_z . Such real-time reaction to fluctuating Hamiltonian parameters can find its application in other qubit encodings and materials, not necessarily limited by nuclear noise in the low-frequency spectrum. As a relevant example, the recent realization of diabatic CPHASE gate between single-electron qubits in silicon [50] (new reference), relies on the precise knowledge of their S-T Hamiltonian parameters, i.e. both ΔB_z , J. Thus, methods of their fast estimation and adaptive control are crucial for achieving its high performance over time.

Regarding our ST qubit setting, working with no deterministic component of dBz poses an additional challenge. This indeed requires setting a control loop that performs the experiment only above a certain threshold of a random field ΔB_z , however this is by no means a trivial operation. Naturally with AC drive of S-T qubit the slow-fluctuations of J, which enters in the qubit energy splitting, result in a low-frequency transverse noise that can still cause both dephasing and phase shift of the Rabi rotations [51,52] (new references).

We agree the presence of the constant ΔB_z (either from nuclear pumping or a micromagnet) would only improve the estimation and feedback control. From this perspective this work is a worst-case scenario. We have modified the manuscript as follows:

Hadamard rotations

Real-time knowledge of both ΔB_z and J would potentially benefit two-qubit gate fidelities [50] and the resonant-driving approach of previous works [35,38,39]. In the resonant implementation, constrained to the operating regime $|\Delta B_z| \gg J$, low-frequency fluctuations of J result in transverse noise that causes dephasing and phase shifts of the Rabi rotations [51, 52].

Discussion

A constant Overhauser field gradient, whether stemming from nuclear spin pumping or a micromagnet, is expected to further improve the feedback control. From this perspective, our work represents a worst-case scenario, demonstrating the effectiveness of our experimental technique.

Comment #4

4. Is this a scalable approach?: I can easily imagine excessive experimental overhead when one attempts to do a multi-qubit experiment with this technique. Not only very big FPGA resources are needed for just a few-qubit control, but also many independent sensors are required since the technique fundamentally relies on fast projective measurement on each qubit. The latter is also a fundamental bottleneck of the scheme: As this is measurement-based feedback control, it is classical feedback, or efficient calibration problem, not generally adaptable while performing coherent operations. Moreover, exponentially increasing waiting time is expected due to the need for simultaneous heralding of meeting the right conditions for all qubit frequencies.

Response: We believe this method shares no more scalability issues than the previously mentioned AC driving and nuclear pumping as they all require fast projective measurement on each qubit. In fact when scalability is concerned, the methods developed in this manuscript could be a useful tool for characterization of random and possibly slowly varying spin-splitting gradients in larger qubit arrays. This includes electron-based devices (or parts of devices) without costly isotopic purifications or occurrence of charge noise sensitive gradients in materials with significant spin-orbit interaction. e.g germanium holes. In general, reliable knowledge of gradients is a crucial element needed for high fidelity readout, as well as for the operation of S-T qubit and two qubit gates between LD qubits (see previous question).

As mentioned in the previous responses, our work is not meant to promote multiple ST qubits operation. At the current stage it is not clear what the future FPGA resources and limitations will be to operate a multi-qubit experiment, and it is outside the scope of this work. We speculate that a large overhead on classical computing and classical FPGA-based quantum control is a valid path forward, as these are relatively easy to engineer. Offloading technological developments from scaling of qubit hardware to scaling of classical control hardware is worth discussing, but not in this manuscript.

Comment #5

5. Using commercially available FPGA, significant improvement in estimation speed?: Overall, I find that the speed of estimation, single-shot readout duration, Bayesian calculation time, and pulse controllability (like granularity) are all comparable to existing experiments using home-built FPGA boards. I do not see significant technical improvement on the electronics side.

Response: We believe that the general-purpose FPGA-powered qubit controller used in our work is state-of-art. It may be possible that homebuilt FPGA boards can compete or exceed the capabilities, but these are likely localized to specific institutions with FPGA domain knowledge or customized to niche applications. The impact of using general-purpose quantum controllers, widely available to the research community, lies in the ease of transfer of our results to other materials and other quantum systems. Over time, control electronics will likely get even better, further improving the results that can be obtained with real-time control. Our manuscript is indeed not intended to demonstrate a hardware improvement, but makes full use of state-of-the-art hardware capabilities.

Response to Reviewer #2:

Comment #0

The authors reported,

- i. Real-time two-axis control of a GaAs S-T0 qubit utilizing FPGA fast feedback control*
- ii. Controlled Overhouser gradient driven rotations*
- iii. Controlled exchange driven rotations*
- iv. Demonstration of Hadamard rotations utilizing the two-axis control*

I think that these are novel and of interest to readers in the community.

However, the following issues should be considered because these might relate to the main point of this paper.

Response: We thank the Reviewer for understanding the novelty of our work. We have modified the manuscript to address all the comments raised by the Reviewer.

Comment #1

Measurement setup

- 1. Indicating the position of $\epsilon = 0$ in Fig. 1a and b will help readers to understand the situation.*

Response: We thank the Reviewer for this suggestion. We have added the position of $\epsilon = 0$ in Fig. 1a and b.

Comment #2

Controlled Overhouser gradient driven rotations and related operations

- 2. As the author mentioned in the Hadamard section, the sign of ΔB_z will not be obtained in this scheme. What is the typical timescale of the change of the sign?*

Response: Any significantly different from zero value of the field is expected to decay to zero on the timescale given by the autocorrelation time of several seconds, in practice many seconds, see also Ref. 41. We have modified the manuscript as follows:

Hadamard rotations

The sign of the gradient may change over long timescales due to nuclear spin diffusion (on the order of a many seconds [41]), but the measurement outcomes of our protocol are expected to be independent of the sign.

Comment #3

- 3. For future complicated gate operations, readers would like to know whether the lack of information on the sign of ΔB_z affects the operations. Can the authors comment on a good method to obtain the sign? Is the method presented in Ref. 50 enough for practical applications in terms of the required time, for example?*

Response: For single-qubit gates in a ST qubit, the sign of ΔB_z is not important, whereas for more complicated gate operations, see reply to Comment #4.

The method presented in [54] (new numbering) would require measuring the relaxation time of the $|up\ down\rangle$ state compared to $|down\ up\rangle$ every time the estimated field is below a sufficiently low frequency. Considering for instance 100 repetitions of initialized $|up\ down\rangle$ and 100 of $|down\ up\rangle$, this requires a few milliseconds for diagnostic measurements (including linear fitting on the FPGA), negligible compared to the sign reversal time of a many seconds mentioned in the response to Comment #2.

For completeness, we emphasize that for multi-qubit control in ST qubits one could combine our scheme with nuclear spin pumping, (i) to avoid small (slow) ΔB_z and (ii) to know the sign. But since our work is not about promoting GaAs-based ST qubits, but rather about achieving two-axis control in general with multiple noise sources, we decided to not include this step. We have modified the manuscript as follows:

Hadamard rotations

The relative sign of Overhauser gradients [...] could be determined following [54] by comparing the relaxation time of the ground state (e.g $|up\ down\rangle$) of ΔB_z with its excited state ($|down\ up\rangle$). Such diagnostic sign-probing cycles on the FPGA should not require more than a few milliseconds, negligible compared to the expected time between sign reversals.

Comment #4

4. For future complicated gate operations (not only for the Hadamard gate), can the authors comment on a scheme that does not depend on the sign of ΔB_z or is robust against the change of the sign?

Response: The sign is not relevant for single-qubit operations, as the x-axis can be defined by the direction of the ΔB_z field. As mentioned in the responses to Comment #2 and 3, the timescale on which the sign changes is much longer than the typical operation time of a few milliseconds. Nevertheless, for two-qubit gates it is the relative sign that matters [53]. In principle in ST qubits the relative sign can be measured by the parity measurement between nearby electrons, however this operation is naturally outside of the scope of this particular work, which concentrates on single-qubit operations. We have modified the manuscript as follows:

Hadamard rotations

The relative sign of Overhauser gradients becomes relevant for multi-qubit experiments [53] [...].

Comment #5

5. The improvement of the Q factor ($Q > 5$, naively $Q \sim 40$) in Hadamard rotations looks much better than those in the cases of Overhauser gradient driven rotations ($Q \geq 7$) and exchange driven rotations ($Q \sim 6$). What is the reason of this better performance especially in the case of the Hadamard rotations compared to the other cases?

Response: The main difference is that to perform the Hadamard gate, two estimated frequencies are used to correct independently fluctuations of the two control axes. The exchange-driven rotations ($Q \sim 6$) are not stable because of high-frequency noise. The controlled Overhauser-driven rotations ($Q \geq 7$) also look stable after the first few oscillations, due to the dominant low-frequency nuclear noise content. We have modified the manuscript as follows:

Hadamard rotations

The resulting Hadamard oscillations are shown in Fig. 5e and fitted with an exponentially decaying sinusoid, indicating a quality factor $Q > 5$.

(According to this naive fit, the amplitude drops to $1/e$ over approximately 40 rotations, although we have not experimentally explored rotation angles beyond 9π .) **In comparison to exchange-controlled rotations from Fig. 4c, the Hadamard rotations are much more stable, which we attribute to the additional feedback on detuning that fixes the oscillation axis and decreases sensitivity to charge noise.**

Response to Reviewer #3:

Comment #0

The manuscript entitled "Real-time two-axis control of a spin qubit" by Berritt et al. discusses the implementation of real-time two-axis control of a qubit with two fluctuating Hamiltonian parameters: the qubit frequency and the J-driven rotations. The proposed method allows for counteracting fluctuations along both axes, resulting in an improved quality factor of coherent qubit rotations. They show that adaptive baseband control pulses can reliably operate a qubit out of slowly fluctuating environments. They also show how they can examine and mitigate low-frequency noise at different operating points of the qubit. The effectiveness of the proposed protocol to stabilize and improve the performance of a singlet-triplet qubit is demonstrated in a series of nice experiments, from the controlled rotation of the ST_0 qubit to real-time universal ST_0 control, with the qubit being stabilized in real time using FPGA-powered techniques.

Overall the paper is very well written and presents an important contribution to the field of the control of spin-based qubits. I believe it is well suited for publication in Nature Communications.

Response: We thank the reviewer for understanding the importance of our work in the field of control of spin-based qubits and thereby recommending publication in *Nature Communications*.

REVIEWERS' COMMENTS

Reviewer #1 (Remarks to the Author):

The authors have made significant efforts to improve the manuscript. Below I briefly put my reply to the replies to the previous comments, followed by my final evaluation.

1. Reply to #1: I understand the point and now generally agree that the content of two-axis estimation has novelty. Remaining concern: see below
2. Reply to #2: I acknowledge the new data. Remaining concern: see below
3. Reply to #3: I agree with the authors' point. No remaining concern
4. Reply to #4: I agree with the authors' point that original comment #4 is an interesting point but out of the scope of this research. No remaining concern
5. Reply to #5: I agree. No remaining concern

Final evaluation

My main remaining concern is that, although I agree with the authors to some extent that the problem is interesting and the work has elements of novelty, the improvement is not dramatic. This concern is common to my points # 1 and # 2. I feel that this reservation will not be resolved, but I at least understand depending on the specific topic, sometimes showing proof-of-concept of the method is more important than showing high-fidelity (or high quantum oscillation quality). With this reservation, I generally recommend the publication.

Reviewer #2 (Remarks to the Author):

I thank the authors for revising the manuscript and addressing my comments.

I have read their point-by-point response and am happy with their answers and revisions.

I recommend the paper for publication.